# Toward Gripper-Integrated Active Electrosense for Pre-Contact Sensing in Underwater Soft Grippers

*Abstract*— **Underwater manipulation often occurs under degraded visibility due to turbidity, glare, and gripper occlusion, limiting the reliability of vision-based perception during approach and grasping. In such settings, soft grippers are well suited for compliant interaction, but they typically lack an onboard pre-contact cue that can guide approach and closure when vision is unreliable. This extended abstract explores active electrosense as a lightweight sensing modality that can provide a proximity-like signal prior to contact by measuring perturbations of an applied electric field in conductive media. We instrument an octopus-inspired gripper with a discrete electrode layout and record multi-channel sensing voltages using off-the-shelf hardware. Simulation and tank experiments with a suspended conductive sphere show structured, object-dependent changes in the multi-electrode voltage readout relative to empty-water baselines, with detectability varying across excitation of 5 to 20 V and frequencies from 1 mHz to 1 kHz. These findings motivate systematic investigation of gripper-integrated electrosense as a complementary pre-contact cue for underwater soft manipulation.**

*Index Terms*— **Active electrosense, underwater manipulation, soft grippers, pre-contact sensing**

## I. INTRODUCTION

Underwater manipulation is often performed under degraded visibility due to turbidity, specular highlights, and gripper self-occlusion, reducing the reliability of optical perception during final approach and closure. This is precisely when the robot must determine whether an object is within the grasp volume, whether the approach should be corrected, and when closure should begin. While tactile sensing can provide rich information after contact has established, it does not provide a pre-contact cue, which is particularly limiting for compliant grippers whose interaction dynamics can change significantly with premature or misaligned contact.

Soft grippers improve underwater grasp robustness through compliance and bio-inspired morphology, and octopus-inspired designs have demonstrated reliable grasping in turbid environments [1], [2]. Once contact is established, dedicated underwater tactile sensors can further improve execution by providing post-contact information such as target recognition, contact state, and force or geometry estimates [3], [4]. However, these sensing modalities remain fundamentally contact-driven and therefore cannot directly support pre-contact approach and closure decisions under poor visibility.

Active electrosense provides a non-contact sensing mechanism in conductive water by applying an electric field and inferring nearby objects from the perturbations they induce [5]. Prior works have demonstrated active electrosense for underwater object detection and identification and for re-active inspection behaviors [6], [7], and pre-touch electric sensing in water has been explicitly introduced and studied [8]. More recent studies have addressed practical deployment issues such as sensor placement and inverse formulations, including sparse reconstruction [9], [10], and electrosense has also been used in system-level localization and mapping pipelines [11]. These studies establish the modality as viable, but they largely treat electrodes as body-mounted arrays or dedicated experimental rigs, rather than sensors integrated into a gripper for manipulation.

This extended abstract addresses that manipulation-centric integration problem. Our goal is not to re-establish that electrosense works in water, but to examine whether a gripper-integrated electrode layout can provide a local, object-dependent multi-electrode signature prior to contact that complements post-contact tactile sensing in underwater soft grasping. We first use simulation to test whether target motion within the grasp region produces structured, position-dependent changes across an electrode array. We then report tank measurements using a reconfigurable gripper skeleton with distributed electrodes and a suspended conductive sphere target, including a coarse sweep over excitation amplitude ($V_{pp}$) and frequency ($f$). Rather than claiming calibrated detection performance, we provide feasibility evidence and highlight dominant design dependencies that control detectability, including excitation regime ($V_{pp}, f$), grounding, and electrode placement.

Our contributions are intentionally preliminary:

- A gripper-integrated electrode configuration and simple drive–sense acquisition pipeline for underwater electrosense measurements.
- Simulation evidence that a conductive target induces structured, position-dependent changes in the multi-electrode readout within the grasp region.
- Tank measurements showing object-dependent signatures and strong $V_{pp}, f$ dependence, motivating systematic follow-up toward robust pre-contact sensing.

The remainder of this extended abstract is organized as follows. Section II reports simulation results on field perturbations and position-dependent signatures. Section III presents underwater experiments, including the setup, single-pair characterization, gripper-level sweep, and preliminary findings. Section IV discusses limitations and next steps. Section V concludes.

## II. PRELIMINARY SIMULATION EVIDENCE

This section uses simulations to establish a manipulation-relevant feasibility property for gripper-integrated elec-

trosense. Specifically, we test whether a conductive target moving within the grasp region produces a spatially structured response across a distributed electrode array, rather than a uniform offset. Such a position-dependent multi-electrode signature is the minimal requirement for gripper-level pre-contact sensing because it provides observability from local measurements within the grasp volume. The simulations are implemented as electrostatic finite-element simulations in COMSOL Multiphysics (COMSOL AB, Stockholm, Sweden).

Fig. 1 summarizes the simulated configuration and the resulting signatures. The model includes a drive–sense electrode layout on an octopus-inspired soft gripper body, where a set of excitation electrodes induce an electric field in conductive water and a ground reference closes the circuit (Fig. 1a–b). We evaluate four representative target positions (Pos 0–Pos 4) by placing a conductive sphere at different locations relative to the gripper and visualizing the resulting electric potential distribution in the grasp volume (Fig. 1c–f). Across positions, the perturbation remains localized around the target but changes the surrounding electric field potential in a position-dependent manner, reshaping the local gradients sampled by nearby sensors.

The key outcome is the predicted sensing-voltage pattern across the sensor index (Fig. 1g). Each target position yields a distinct multi-electrode signature characterized by non-uniform, multi-peak responses whose relative amplitudes and ordering change with object position. This indicates that the array response encodes coarse information about target position within the grasp region and supports the use of multi-channel readouts and per-electrode patterns, rather than a single scalar measurement, in subsequent experiments. In the remainder of the this extended abstract we therefore report both per-sensor signatures and an aggregate separation score to summarize detectability under different $V_{pp}, f$ settings. For clarity, the electrode layout in this simulation is a representative configuration used to study position-dependent field perturbations; the physical electrode placement used in

the tank experiments differs due to hardware and prototyping constraints and is described in the experimental section.

## III. UNDERWATER EXPERIMENTS AND PRELIMINARY RESULTS

### A. Experimental setup

All experiments use a drive–sense measurement architecture (Fig. 2d). A signal generator applies a square-wave excitation to the designated excitation electrodes to generate an electric field in water, while the remaining electrodes are recorded as sensing channels. The sensing voltages are acquired simultaneously using an NI USB-6210 DAQ and logged on a host PC for baseline subtraction and subsequent analysis. Two physical configurations are evaluated using the same excitation and acquisition chain: a minimal sender–receiver fixture for pairwise characterization (Fig. 2a, inset) and a reconfigurable multi-electrode gripper prototype for gripper-level measurements (Fig. 2b–c).

### B. Single-pair characterization

We first characterize a minimal sender–receiver fixture to identify frequency regimes that yield a pronounced received signal under the present hardware and water conditions (Fig. 2a). The fixture geometry is intentionally simplified and does not match any specific electrode pair on the gripper; its role is to isolate frequency dependence of the measurement chain and electrode–water interaction before introducing the multi-electrode gripper geometry. Using this simple setup, we sweep $(f)$ from 1 mHz to 1 MHz and record the receiver voltage magnitude $V_{\text{sense}}$ for two $V_{pp}$, 2 V and 10 V (Fig. 2a). The response varies substantially across the sweep, indicating that $f$ is a primary design parameter for practical electrosense. The two $V_{pp}$ provide a low-amplitude reference and a higher-amplitude condition that remains compatible with the USB-6210 input range, allowing a coarse check of signal scaling without requiring a full amplitude characterization. The gripper-level experiments reported next therefore treat $V_{pp}$ and $f$ as design variables and explicitly explore a broader $V_{pp}$ range.

### C. Gripper-Integrated Experiments

We next evaluate gripper-integrated electrosense using the same drive–sense instrumentation chain (Fig. 2d). To enable rapid iteration of electrode placement and wiring during early-stage testing, we conduct the tank trials with a reconfigurable ribbed skeleton that preserves the grasp-region geometry while allowing sensors to be repositioned (Fig. 2b). A conductive metal sphere is used as a simple target and is positioned 10 cm above the grasp region to focus on pre-contact sensing (Fig. 2c). For each $V_{pp}, f$ setting, we obtain an empty-water baseline and an object-present reading with the same nominal gripper pose and target placement.

At each trial, the multi-electrode readout is represented by a measurement vector $\mathbf{v} \in \mathbb{R}^m$ ($m$ denotes number of sensing channels), whose entries are the simultaneously recorded sensing-channel voltages. Object-induced changes

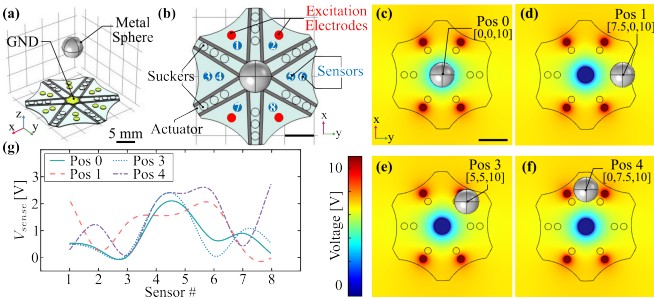

Fig. 1. **Simulation study of gripper-integrated electrosense and object-position dependence. (a)** 3D simulation setup showing an octopus-inspired gripper with a conductive metal sphere target and ground electrode. **(b)** Electrode layout used in simulation, indicating excitation electrodes (red) and sensors (blue) distributed around the grasp region. **(c–f)** Example electric potential slices for different target positions (Pos 0–Pos 4), illustrating how the field distribution in the grasp volume is perturbed as the conductive sphere moves relative to the gripper. **(g)** Corresponding time-averaged sensing voltage patterns across the sensor set for each target position, demonstrating that the multi-electrode signature varies with object pose.

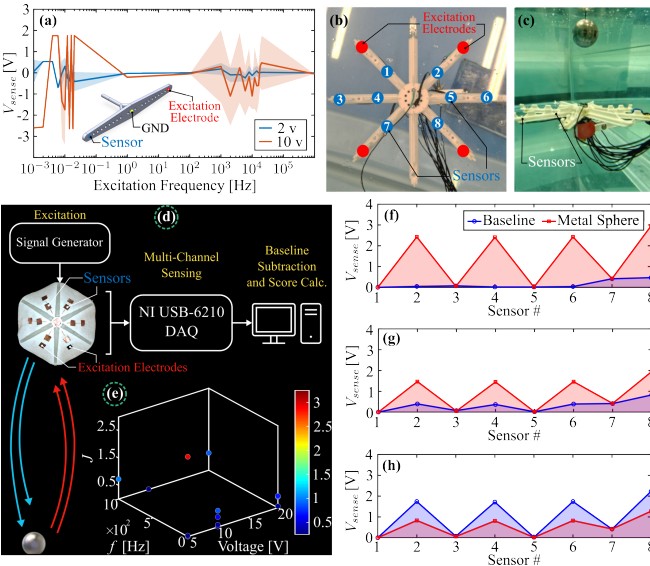

Fig. 2. **Experimental pipeline and preliminary measurements for underwater gripper-level electrosense. (a)** Minimal sender–receiver characterization showing receiver voltage versus $f$ for two $V_{pp}$. **(b)** Reconfigurable gripper skeleton used to prototype electrode placement (excitation in red, sensing in blue). **(c)** Tank experiment configuration for pre-contact sensing with a suspended conductive sphere. (d) Instrumentation pipeline: signal generator excitation, multi-channel acquisition using an NI USB-6210 DAQ, and baseline subtraction with $J(V_{pp}, f)$ computation. (e) Separation score $J(V_{pp}, f)$ over a coarse grid of $V_{pp}, f$. (f–h) Representative per-sensor voltage patterns comparing empty-water baseline and metal-sphere conditions: (f) $V_{pp} = 5$ V, $f = 1$ mHz; (g) $V_{pp} = 10$ V, $f = 100$ Hz; (h) $V_{pp} = 20$ V, $f = 1$ kHz.

are summarized by pairing baseline and object-present measurements at the same excitation setting and computing the separation score

$$J(V_{pp}, f) = \|\mathbf{v}_{obj}(V_{pp}, f) - \mathbf{v}_{empty}(V_{pp}, f)\|_2, \quad (1)$$

where $\mathbf{v}_{empty}$ and $\mathbf{v}_{obj}$ denote the empty-water and object-present voltage readouts, respectively. We use $J(V_{pp}, f)$ as a coarse indicator of detectability across excitation regimes and interpret it alongside per-sensor signatures.

We perform a coarse parameter sweep over $V_{pp} \in \{5, 10, 20\}$ V and frequency $f \in \{1\,\text{mHz}, 1\,\text{Hz}, 50\,\text{Hz}, 100\,\text{Hz}, 1\,\text{kHz}\}$ to map how detectability varies with excitation settings rather than to identify a globally optimal operating point. Fig. 2e reports the resulting $J(V_{pp}, f)$ summary, where larger values indicate a larger aggregate change between baseline and object-present trials at a given grid point. In addition, Figs. 2f–h show representative per-sensor voltage patterns $V_{\text{sense}}$ across the sensor index for baseline and metal-sphere conditions under three excitation settings, $V_{pp} = 5$ V at $f = 1$ mHz, $V_{pp} = 10$ V at $f = 100$ Hz, and $V_{pp} = 20$ V at $f = 1$ kHz. These examples show that object presence is expressed primarily as a spatial pattern change across sensors, and that both the overall amplitude level and the baseline-to-object contrast vary with excitation settings, motivating joint reporting of per-sensor signatures and the aggregate $J(V_{pp}, f)$.

## IV. Discussion, Limitations, and Next Steps

The gripper-level results indicate that object presence is expressed primarily as a spatial signature across sensors rather than as a uniform offset. The per-sensor plots in Fig. 2f–h show a repeatable multi-peak pattern in the object-present condition, whereas the empty-water baseline remains comparatively low. This observation is relevant for manipulation because it supports a local, multi-channel cue that can be evaluated prior to contact, instead of relying on contact-driven sensing alone.

The experiments also show strong dependence on excitation settings. Both the aggregate $J(V_{pp}, f)$ in Fig. 2e and the representative per-sensor signatures indicate that baseline level and baseline-to-object contrast vary with $(V_{pp}, f)$. This suggests that excitation design and grounding are not secondary implementation details, but primary determinants of detectability for gripper-integrated electrosense. A plausible explanation is that different excitation regimes alter the effective current paths through the conductive medium and the target, while low-frequency operation may be more susceptible to slow drift and electrode–electrolyte interface polarization. These factors define a practical design space that must be characterized to make electrosense reliable for underwater grasping.

### A. Limitations

This extended abstract is intentionally scoped. The evaluation uses a single conductive target at a fixed distance in tank experiments, and we do not report comprehensive repeatability statistics. The acquisition and processing are deliberately lightweight, relying on baseline subtraction and a simple $J(V_{pp}, f)$ without optimized analog conditioning or demodulation. The simulations are qualitative and do not model electrode–electrolyte interface polarization, wiring and grounding parasitics, or bandwidth limitations of the acquisition electronics. These constraints limit our claims to feasibility and trend visualization rather than calibrated performance.

### B. Future Work

Future work should prioritize controlled distance sweeps and repeated trials to quantify sensitivity and variability, together with explicit characterization of water conductivity and grounding effects. On the excitation and processing side, waveform design and demodulation, including lock-in style processing, can better isolate informative components of the response while suppressing drift and environmental noise. Finally, the manipulation-relevant milestone is closed-loop integration, where electrosense provides a pre-contact cue for approach correction or closure timing prior to contact and is then complemented by contact-based sensing during grasp execution.

## V. Conclusion

We presented early feasibility evidence for gripper-integrated active electrosense as a pre-contact cue for underwater soft manipulation. Simulation and tank experi-

ments with an octopus-inspired, electrode-instrumented gripper show that a conductive sphere induces a structured, object-dependent change in the multi-electrode voltage readout relative to empty-water baselines. A coarse sweep over excitation amplitude and frequency reveals strong excitation regime dependence in detectability, indicating that excitation settings, electrode placement, grounding, and processing choices are primary determinants of performance. These results motivate systematic follow-up studies toward robust gripper-level pre-contact electrosense for underwater soft grasping, including improved excitation and processing, environmental robustness tests, and closed-loop integration.

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
