# OpenReview forum: "Toward Gripper-Integrated Active Electrosense for Pre-Contact Sensing in Underwater Soft Grippers"
_IEEE.org/ICRA/2026/Workshop/Manipulation_Robustness — ICRA 2026_

### Official Review · Reviewer_zAc4 · 2026-05-09

**Rating:** 7
**Confidence:** 4

**Review:**

**Summary:**
This paper presents electrosensing using discrete electrodes on a soft octopus style gripper to show initial feasibility and proof of concept of pre-contact object presence detection. Perturbations of an applied electric field are measured underwater using multi-electrode readouts. Initial simulation results are constructed using COMSOL (a finite element simulation platform) to validate position-dependent field perturbations prior to real-world experiments.

**Strengths:**
The paper is clearly motivated with initial real world and simulation results, showing the real tractability of electrosensing. Underwater systems have high uncertainty. The authors are honest with their scope and choose a practical prototyping platform. The simulator of choice is also well respected in mechanical engineering communities for accuracy and high resolution for complex simulation. The authors do have a sim to real gap, see weaknesses.

**Weaknesses:**
The method only works in conductive mediums, without characterization of how conductive a medium needs to be for electrosensing to work well. For example, does it work equally well in both fresh and salt water environments?

The paper uses a metal sphere as the object of choice for grasping, which does not account for variations in object shape, size, or general variation. Additionally, it is unclear how sensitive the method is to object conductivity, as real-world targets may be non-conductive. Further testing is needed to ensure true generalization.

The electrode layouts used in simulation and the physical gripper do not match due to hardware constraints, weakening the ability to use COMSOL as a predictive design tool for the real system. The separation score J is computed as a raw L2 norm, meaning a noisy high-amplitude setting and a clean low-amplitude one could score similarly, making it a potentially misleading detectability metric without normalization or explicit SNR reporting.

The paper is primarily from a mechanical background; I would recommend adding a brief explanation for how exactly electrosensing works with examples for non-hardware audiences and introducing COMSOL briefly. The paper could also be strengthened by adding a title figure and an extended explanation for why current underwater methods (i.e. vision etc.) are not sufficient.

---

### Decision · Program_Chairs · 2026-05-21

Accept